# Mental Health and Quality of Life among University Students with Disabilities: The Moderating Role of Religiosity and Social Connectedness

Ebrahim A. Al-Shaer [1], Meqbel M. Aliedan [1], Mohamed A. Zayed [2,3,*], Musaddag Elrayah [1] and Mohamed A. Moustafa [1]

1   Department of Management, School of Business, King Faisal University, P.O. Box 400, Al-Hassa 31982, Saudi Arabia; ielshaer@kfu.edu.sa (E.A.A.-S.); maliedan@kfu.edu.sa (M.M.A.); melrayah@kfu.edu.sa (M.E.); mamoustafa@kfu.edu.sa (M.A.M.)
2   Department of Fitness, Gymnastics and Sports Show, Faculty of Physical Education Alexandria University, Alexandria 21625, Egypt
3   Deanship of Student Affairs, King Faisal University, Al-Hassa 31982, Saudi Arabia
*   Correspondence: mazayed@kfu.edu.sa

**Abstract:** This research paper explores the intricate interplay between mental health (MH), quality of life (QOL), religiosity, and social connectedness among students with disabilities. In the context of a growing awareness of the multifaceted nature of well-being, this study aims to unravel the moderating effects of religiosity and social connectedness on the relationship between mental health and the overall quality of life experienced by students with disabilities. Utilizing Partial Least Squares Structural Equation Modeling (PLS-SEM), this research investigates direct and moderating effects within the proposed conceptual framework. The results indicate that all mental health disorder dimensions (stress, depression, and anxiety) negatively and significantly affect the quality of life of students with disabilities; moreover, the negative impact of stress on QOL is dampened by the moderation effect of social connectedness. Similarly, the negative impact of anxiety on QOL is dampened by the moderation effect of religiosity. However, the findings indicate that social connectedness fails to dampen the negative impact of depression (and anxiety ($\beta = -0.12$)) on QOL. Similarly, the findings reveal that religiosity is unable to dampen the negative effect of depression and stress on QOL. These findings' implications extend to the theoretical and practical domains, informing interventions and support systems aimed at enhancing the overall quality of life of people with disabilities.

**Keywords:** sustainable; stress; depression; anxiety; mental health; social connectedness; quality of life; religiosity

## 1. Introduction

Today, many universities around the globe have developed sustainable admissions policies regarding students with disabilities. The sustainable development goal (SDG3) clearly states the importance of mental health as crucial in people's overall health and well-being, aiming to guarantee everyone's health and well-being at every stage of life [1]. Sustainable development may be impacted by people's inability to contribute to society and the environment due to inadequate physical or mental health. Mental health and its various aspects, such as emotional, social, financial, and physical health, are currently at the forefront of global public health issues [2]. Older students have a lower chance of any disorder in every institution than younger ones [3,4].

Worldwide, there are now more pupils with impairments than ever before. When compared to their peers without disabilities, students with disabilities frequently face challenges in educational settings, which leads to gaps in their academic performance and

worse results in terms of college and employment possibilities [5]. Approximately 11% of undergraduate students disclose that they possess a disability or impairment which can impact students' sustainable ability to engage in major life and academic activities. These difficulties, such as mental health (MH), affect students' quality of life (QOL) [6].

QOL is defined as a person's evaluation of their role in the community, considering the culture, norms, and values surrounding them. This includes their goals, perspectives, standards, and concerns [7]. For example, medical students face a heightened likelihood of experiencing depression throughout their training, thus establishing a connection between events promoting MH and well-being, and the provision of accommodations for individuals with disabilities becomes crucial [8].

Various factors, including support from peers, services specifically designed for individuals with disabilities, teaching methods employed by faculty, the overall campus environment, and the ability to advocate for oneself, are indicative of a high level of academic degrees among disabled college students [9].

Students' QOL is very crucial for academic success [10,11]. Students with psychiatric disabilities derive an understanding of their disability by performing interaction activities with colleagues and showing active participation in a range of activities and experiences, including non-class activities [12].

In addition, MH is considered to be an important part of students' QOL [13]. University students' MH and quality of life (QOL) have become more of a concern worldwide [14]. Many factors affect the MH and QOL of students with disabilities. These factors include the main components of MH symptoms like depression, anxiety, and stress [15–17]. These factors negatively impact students with disabilities in terms of their MH and QOL [18,19]. In addition, environmental difficulties, such as physical access to buildings and curriculum issues including teaching, learning, and evaluation, frequently pose challenges to students with disabilities [20]. We can argue that sustainable university practices might improve the QOL of all students, including those with disabilities. Environmental issues are very important to the lives of students.

During the past and in recent years, the issue of MH and its relation to academic performance and QOL among students has been studied by many researchers [21,22]. Students' own beliefs might affect their MH status. Many factors might improve MH quality as well as QOL for all students. In a study conducted by Alireza Maredpour, the significance of belief in a higher power was underscored, as well as social and spiritual assistance and a feeling of belonging connected to a superior entity, in upholding the optimal psychological conditions when confronted with distressing circumstances [23].

This study aims to find out how MH relates to QOL. It tries to add value to the current literature by focusing mainly on the moderating role of religiosity and social connectedness in the relationship between the MH and QOL of students with disabilities. The study tries to answer the main questions, like how MH components—depression, anxiety, and stress—affect students with disabilities? Do religiosity and social connectedness moderate the relationship between MH and QOL?

Because it focuses on impaired university students, this research is significant. Students encounter various situations that negatively affect their MH and QOL. This study attempts to draw attention to the anticipated roles that social connectivity and religiosity will play in the mental health of impaired students—that is, their levels of stress, anxiety, depression, and QOL [24].

Different studies have found that social support, particularly the actual receipt of support by students with disabilities, is linked to improved MH [25,26]. Religiosity pertains to the convictions and conduct of individuals that tackle profound or surpassing matters in the scholarly investigation of faith. The phenomenon encompasses a variety of aspects, including diverse dimensions, religious orientations of different types, and varied religious experiences. It additionally encompasses the symbolic and meaning-enhancing roles played by religious devotion and the organization and functioning of national sentiment.

In recent times, a form of faith has emerged centered on individual interpretations and personal selections. This development introduces fresh obstacles in the examination of religiosity in the current era [27]. Hence, the presence of religiousness and the establishment of social connections might have the potential to enhance the psychological welfare and overall life satisfaction of disabled individuals pursuing higher education by imbuing them with a sense of purpose, importance, and companionship in times of adversity [28].

This research tries to fill the gap in the current literature, especially MH and it relationship with the QOL among students with disabilities. We try to highlight the moderation impact of social connectedness and religiosity. Very few studies have explored this moderation impact.

In addition, this research is expected to have educational, social, health, and economic implications. Very few studies have focused on the role played by the main components of MH among students with disabilities who expect to make positive contributions to their community and support the SDG3. Thus, we try to explore how these factors affect QOL, taking into consideration the intervention of religiosity and social connectedness as moderators in this relationship.

Sustainability, often associated with environmental concerns, has a multidimensional impact on societal health. In the context of mental health and quality of life among university students with disabilities, sustainability extends beyond ecological aspects to encompass social, economic, and personal dimensions. Fostering a supportive and inclusive university environment for students with disabilities contributes to social sustainability. This involves creating a community that values diversity, promotes equal opportunities, and ensures the well-being of all individuals. Economic aspects are intertwined with sustainability, and in the context of university life, this involves providing financial support, resources, and accessible facilities for students with disabilities. This support contributes to economic sustainability by enabling equal access to education and resources. Sustainable practices in a university setting also include prioritizing students' mental health and well-being. Initiatives that address the unique challenges faced by students with disabilities, such as the moderating role of religiosity and social connectedness, contribute to personal sustainability and quality of life. Sustainability emphasizes resilience and adaptability in the face of challenges. This study focuses on the moderating role of religiosity and social connectedness and suggests that these factors contribute to the resilience of students, helping them yo navigate the university environment more effectively.

## 2. Literature Review

### 2.1. Depression and the QOL

It is acknowledged that depression is a severe public health problem that has a substantial negative impact on the personal, financial, and social well-being of persons who experience it, as well as their family [29]. It is a prevalent psychological condition that includes a mood of sadness, a lack of enthusiasm or enjoyment, a sense of guilt, or a lack of worth [30]. It is more common in students with learning problems than in those who do not have a disability related to learning. It represents a prevailing global health concern, serving as a prevalent catalyst for disability and impacting approximately 9% of males and 17% of females residing in Europe [31]. It may have an impact on various aspects of the lives of students with disabilities, such as their academic performance, social relationships, physical health, and overall well-being. It may give rise to challenges in the area of concentration, issues with memory, and a decrease in motivation, thereby potentially culminating in substandard academic achievements. Teenagers who have disabilities are more likely to have emotional problems as a result of depression, such as sadness, anxiety, low self-esteem, and loneliness, which can affect their QOL and academic grades, as well as their general performance. It has been discovered that students with disabilities perform noticeably worse academically than their peers, particularly in the absence of complete disability support services [32]. There is a growing prevalence of MH issues among students globally, and several scholars have contended that this can be linked to the pressures of academia,

financial burdens, and social stressors [33,34]. A deficiency in social assistance serves as a determining factor for mental health issues, which encompass symptoms of depression, particularly within the demographic of university students [35,36].

Prior research has revealed a degree of negative correlation between depression severity and subpar academic achievement, as well as a negative perception of QOL among students [37]. Depression has been shown in studies to have a negative impact on QOL among students [38,39]. Students who exhibited depression symptoms lost significantly more healthy periods than those without [40]. The effects of depression and other psychological disorders on students' existing behavior, grades, and QOL are severe. University students who show at least one symptom similar to depression can be described as being more likely to experience a risk of academic difficulties, such as receiving a lower grade, than those who do not report symptoms [41]. People with high levels of depression show a poor QOL [42]. This result indicates how depression can negatively impact people, as well as the QOL of students with disabilities. We can argue that depressed students have a poorer QOL in specific situations. Many factors might lead the depression situation to become worse. These factors might include personal, social, economic, and environmental factors. Researchers can develop different models to understand these factors and provide recommendations accordingly.

**H1.** *Depression negatively affects quality of life.*

### 2.2. Anxiety and the QOL

Anxiety is a famous symptom around the globe. In the most current World Health Organization (WHO) report, it was included as one of the most common symptoms among young people [43]. It reflects high levels of excessive fear, anxiousness, apprehension, and accompanying behavioral problems and disorders [44]. Anxiety disorders and sadness are quite common and are linked to lower levels of functioning [45]. Symptoms of anxiety can develop as a result of a mix of genetic, environmental, and mental variables. Previous studies have found that anxiety causes an increase in sadness and nervousness, which leads to a deterioration in academic performance among students [46]. The most important factors that are thought to have negative consequences and implications on QOL are the conditions of happiness, success, and wealth, which are affected by level of anxiety [47]. Anxiety disorders were discovered to cause a lower QOL [48]. Level of anxiety was found to be related to poor academic performance and a low QOL [49]. Anxiety among disabled people was found to be increased and affect their QOL [50]. Previous research has consistently demonstrated that girls are more worried than boys [51]. A low QOL and increased disease activity have been associated with anxiety [52,53]. Anxiety disorders have been linked to a lower QOL in a variety of categories, including physical, psychological, social, and environmental aspects [54]. Students with higher levels of anxiety score lower in a variety of QOL dimensions, including physical, psychological, and social interactions, and the environment. Anxiety has a direct negative impact on QOL, and academic stress has a direct poor impact as well [55]. Students' encounters with more recent traumatic incidents are also significantly related to low retention and grades at university [56]. Previous research has discovered a link between anxiety levels and QOL. The majority of these studies have shown negative consequences of stress [54,57,58]. Students with disabilities might be affected more than their peers by level of anxiety. Thus, universities around the globe have launched specialized support units for students with disabilities to improve their QOL.

**H2.** *Anxiety negatively affects quality of life.*

### 2.3. Stress and the QOL

Stress among students is a general symptom. According to the WHO, students need to be physically and psychologically comfortable to fully engage in education [59]. Stress

among students, especially those with disabilities, negatively impacts their academic life, as well as their QOL. Previous research has linked self-reported stress to the manifestation of anxious moods and poor mental health [60]. This might also affect students' engagement during class activities. The frequency of good feelings during class is connected to better student involvement and participation. Negative emotions, on the other hand, are connected to a lesser participation level [61]. According to the aforementioned OECD poll, senior students described as having higher levels of academic-related stress developed a poorer QOL [62]. This might have occurred because of their academic load at this level of education. Continuous orientation and advising can control stress levels at all academic levels. The emotions experienced during educational activities have been linked to critical outcomes such as academic adaptability and success, as well as student well-being and mental health. Stress and worry, on the other hand, can contribute to poor academic performance and sickness. This means that stress creates a negative perception of students with disabilities about their QOL [63]. Stress has a negative association with QOL. Students with disabilities show high levels of stress, which translate into low grades as well as a poor QOL [64]. University students in the United States have been demonstrated to experience less sleep as a result of self-perceived stress [65]. Continuous stress can reduce impaired students' overall QOL by lowering their general life satisfaction, sense of self-worth, and overall well-being. The authors found a negative relationship between level of stress and QOL [66]. It can be described as negatively predicting quality of life [67].

**H3.** *Stress negatively affects quality of life.*

### 2.4. The Moderated Role of Religiosity and Social Connectedness

The degree to which people hold religious concepts, practices, and activities has been shown to have a moderating effect on several aspects of MH, including depression. Religion can also influence the link between depression and QOL [68]. Religious ideas and practices frequently offer individuals religiously appropriate coping methods [69,70]. Inner faith is connected to resilience, QOL, and fewer previous suicide attempts. As a result, it contributes to a better QOL [70]. The extent to which religious involvement influences the relationship between depression and quality of life (QOL) among male and female students with disabilities has not been thoroughly investigated. We anticipate that the presence of religious beliefs will exert a beneficial moderating effect, mitigating the adverse consequences of stress, anxiety, and depression on one's overall well-being. Adopting religious practices can enhance mental health (MH) by providing individuals with a sense of purpose and direction in life. Conversely, religion does not appear to have any bearing on the association between stress and QOL [71]. In addition, social connections as well as campus connections can help to improve the negative impact of depression, anxiety, and stress on QOL. Students have reported that meaningful interactions with family, colleagues, friends, or teachers improve their QOL [72].

**H4.** *Social connectedness moderates the relationship between depression and quality of life.*

Social interaction is critical to one's standard of living and QOL. Since humans are social creatures by nature, deep relationships and meaningful interactions with other people are crucial to our wellbeing in a variety of ways [73]. Social engagement boosts mental health, lessens perceptions of being alone or isolated, and offers emotional support. Reducing stress and anxiety and promoting pleasant emotions can be achieved via conversing, exchanging experiences, and being a part of a community. It might lead to enhancing the relationship between depression, stress, and QOL among students with disabilities [74].

**H5.** *Social connectedness moderates the relationship between anxiety and quality of life.*

Higher levels of pleasure and life satisfaction are associated with social engagement. A greater QOL results from having social support networks and significant relationships.

It offers a sense of contentment, direction, and a safety net to navigate through life's ups and downs. Social connection is critical to a person's standard of living and overall quality of life [75]. It impacts one's physical, mental, and emotional health, sense of belonging, cognitive stimulation, personal development, and general happiness. For this reason, leading a happy and meaningful life requires cultivating social relationships and keeping an active social life. We argue that a high level of social interaction might enhance the people's levels of stress and anxiety. Students with stronger parental relationships and higher levels of religiosity have better mental health than students with weaker parental relationships and lower levels of religiosity [76,77].

**H6.** *Social connectedness moderates the relationship between stress and quality of life.*

Though not with overall depressed symptoms or quality of life, positive religious coping is substantially linked to the positive domains of outcome measures like positive effect and life satisfaction. There is a slight correlation between more religiosity and fewer symptoms [78]. Finding benefits completely mediated the association between positive religious coping and the positive sub-domains of psychological outcomes, according to tests of mediation studies [79]. We can argue that social connectedness might improve QOL and thus improve the relationship between stress and QOL as seen in Scheme 1. We can argue that religiosity is very important for young people and might improve their perception of their QOL. Many studies have studied the moderated role of religiosity in different factors related to life satisfaction [80]. This study is unique and can add value to the current literature by highlighting the role of improvement expected by religiosity.

**H7.** *Religiosity moderates the relationship between depression and quality of life.*

**H8.** *Religiosity moderates the relationship between anxiety and quality of life.*

**H9.** *Religiosity moderates the relationship between stress and quality of life.*

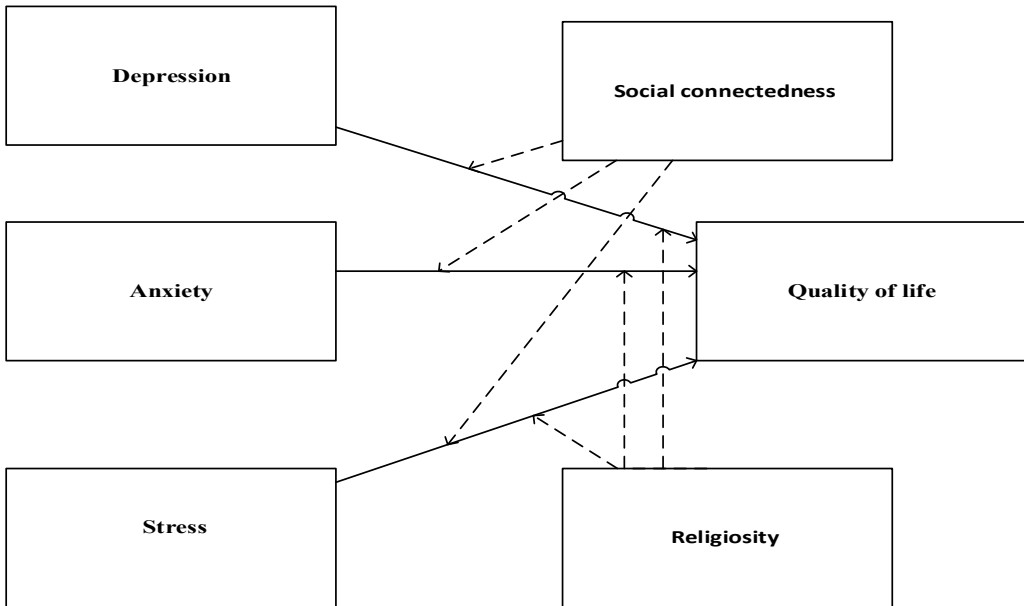

**Scheme 1.** Conceptual Research Model.

## 3. Materials and Methods

A cross-sectional quantitative method was employed to collect and analyze the data. The main aim was to explore the impacts of mental health disorders on the QOL of students with disabilities, and test UDSS and religiosity as moderators. The required information

was acquired through designing an online questionnaire. The survey was structured to include two different parts. There was an introductory part (i.e., age, disability type, gender, and education level), and the second was structured to gather data regarding the study constructs. To operationalize mental health disorders, we used the DASS-21 well-known scale, which assesses signs of mental health disorders (i.e., stress, anxiety, and depression). The DASS-21 is a validated tool that has been used extensively to measure stress, anxiety, and depression [81]. The DASS-21 scale is adjustable for various purposes in various situations and environments. The DASS-21 scale was constructed to assess negative feelings over the prior 7 days, and each factor of the main dimension has seven variables. Students with disabilities were required to express their level of agreement using four-point Likert scoring, where 0 was "no agreement" and 3 was "high level of agreement" [82]. The DASS-21 scale demonstrated adequate consistency and reliability: Depression ($\alpha = 0.965$); Anxiety $\alpha = 0.956$; and Stress ($\alpha = 0.948$). Furthermore "Satisfaction with Life Scale" [SWLS], a scale developed by Diener et al. in 1985, was employed to operationalize QOL [83]. The SWLS explores overall life satisfaction. The participants were required to indicate their level of agreement with various happiness statements such as: "My life circumstances are excellent" and "I wouldn't change much if I could relive my life". Each variable was measured on a five-level Likert, where 1 was "strong disagreement" and 5 was "strong agreement". The SWLS scale showed a satisfactory internal consistency with an $\alpha$ value equal to 0.938. Additionally, religiosity was operationalized using a four-item scale derived from Lombardi, Gerdes, and Murray [84], and the items on this scale exhibited strong internal consistency ($\alpha = 0.946$). Moreover, social connectedness was operationalized with seven items adopted and modified from the Social Connectedness and the Social Assurance Scales introduced by [31]. Example items include: "I feel more comfortable when someone from my family is constantly with me" and "My life is incomplete without a buddy beside me". The items on the social connectedness scale exhibited strong internal consistency ($\alpha = 0.987$). Lastly, religiosity was measured with 7 items, as suggested by Lombardi et al. [84]. Two example statements are: "My faith permeates my entire existence" and "I experience God's presence in my life". QOL and religiosity were evaluated through asking the participant to express their agreement level using a scale from 1 "indicating the lowest level of agreement" to 7 "indicating the highest level of agreement". The internal consistency value ($\alpha$) was 0.975, indicating a strong internal consistency of the utilized scale.

The questionnaires were sent to participants through social network sites (SNS) and university emails during April and May 2023. We used a non-probability convenience sampling technique to ensure a diverse representation of participants. In the questionnaire introduction, we made strictly clear that participant confidentiality and privacy would be maintained and that the collected data would be used for scientific purposes only. To ensure participant safety and prevent confounding variables, individuals who were not currently enrolled as university students were excluded; additionally, individuals unable to sign informed consent were excluded as well. In total, we received 410 replies, and after a precise review of these replies, 390 of them were regarded as valid and 20 questionnaires were eliminated due to incomplete answers, yielding a response rate of 95%.

The disability rate in the country stands at 7.1%, representing 1,445,723 individuals among the total population of 32.94 million. Within the 27 public universities, there are a total of 1.87 million registered students, among whom, 3965 students are identified as having disabilities. The study encompasses all students with disabilities within universities in KSA as its population. Adhering to Roussel's recommendation [85], a sample size exceeding 5 times the items, preferably 10 times or more, was deemed as suitable for a robust data analysis. With the instrument comprising 39 items, the study received 390 complete and valid responses for analysis. The sample size adheres to acceptable standards, as advised by Krejcie and Morgan [86] and Taherdoost [87].

The grouping of gender among participants is fairly even, with 56% identifying as male and 44% as female, as shown in Table 1.

**Table 1.** Respondents' profile.

| Category | Group | Frequency (N = 390) | % |
|---|---|---|---|
| Gender | Male | 218 | 56.0 |
|  | Female | 172 | 44.0 |
| Age group | 17–20 | 156 | 40.0 |
|  | 21–23 | 195 | 50.0 |
|  | 24–25 | 39 | 10.0 |
| study year | Frist year | 68 | 17.4 |
|  | Second year | 103 | 26.4 |
|  | Third year | 177 | 45.5 |
|  | Fourth year | 42 | 10.7 |
| Disability | Physical disability | 129 | 33 |
|  | Visual disability | 137 | 35 |
|  | Hearing disability | 66 | 17 |
|  | Speech disability | 58 | 15 |
| Academic discipline | Art | 156 | 40 |
|  | Education | 137 | 35 |
|  | Business Administration | 58 | 15 |
|  | Agricultural and Food Sciences | 39 | 10 |

The age distribution is diverse, with a notable 60% falling within the 21–25 age group, indicating a significant presence of university students with disabilities in the typical college-age range. Additionally, a substantial 30% of respondents were under or equal to 20 years old, highlighting the inclusion of younger students (first and second year) in the study. The distribution of disability types reflects the varied nature of disability among the respondents, with visual impairment being the most prevalent at 35%, followed closely by physical (mobility) impairment at 33%. This diversity in disability types is crucial for evaluating how different conditions may impact quality of life. As shown in Table 1, the majority of participants belonged to the arts and education disciplines.

*3.1. Dealing with Missing Value, CMV, and Non-Response Bias*

To address missing values in our dataset, we conducted Little's MCAR test using the "Expectation Maximization Algorithm" (EMA) procedure. The results, with a Chi-Square value of 168.770, 157 degrees of freedom, and a significance level of 0.367, indicate that the missing data can be considered as occurring at random. Following this assessment, we implemented the EMA "Expectation Maximization Algorithm", effectively leveraging its imputation capabilities to handle missing values in a manner consistent with the observed random pattern. This approach ensures the integrity and completeness of our data, enhancing the reliability of subsequent analyses and interpretations.

Additionally, in the realm of social and behavioral sciences, the presence of "common method variance" (CMV) within measurement scales can pose a potential threat to the validity of research findings [88]. This phenomenon has the capacity to impact item reliability, alter the covariation among latent constructs, and consequently influence structural relationships [89,90]. To address this concern, ref. [91] recommends procedural design and statistical control. Consistent with the guidance of Podsakoff et al. [88], this study proactively addressed CMV during the questionnaire design phase by implementing strategies such as avoiding common answer effects, mitigating biases in responses, and carefully managing various aspects of scale presentation. Furthermore, an assessment using Harman's one-factor test was conducted, providing reassurance that CMV did not significantly impact the outcomes of this study.

Finally, acknowledging and actively managing non-response bias are pivotal considerations for researchers, as neglecting this issue can compromise the broader applicability and generalizability of research findings [92]. Non-response bias, characterized by variations between respondents and those who choose not to participate, presents a challenge in rep-

resenting the perspectives of non-participants [93,94]. While addressing nonparticipation complexities, researchers are compelled to actively tackle non-response bias to uphold the validity of their studies [95]. Various methods, such as analyzing identified characteristics, examining non-responder subsamples, wave analysis, and linear extrapolation, are commonly employed to assess non-response bias [96]. In this study, a thorough three-step approach, based on the continuum-of-resistance theory [97], was executed to confirm that non-response bias did not significantly impact the findings. First, an analysis of the known demographic characteristics revealed no significant differences between groups, providing confidence in the comparability of the respondents. Additionally, a comparison of respondents from different universities ensured the similarity of samples obtained through two data collection modes, with no significant differences observed. Second, a wave analysis was conducted, scrutinizing variables between early and late responders, reinforcing the robustness of the study's findings against potential non-response bias.

### 3.2. Model Evaluation

In this study, the researchers employed "Structural Equation Modeling (SEM) Partial Least Squares (PLS)" to test the study hypotheses. PLS-SEM is a method that is well-suited for modeling complex cause-and-effect interrelationships, particularly when dealing with multiple latent constructs indirectly measured by several indicators [98]. PLS-SEM serves as an alternative to the conventional covariance-based SEM (CB-SEM) [71]. Widely recognized for its applicability in prediction-oriented and exploratory research, PLS-SEM has garnered significant acknowledgment [99,100]. Operating without the assumption of normality in the sampling distribution, it has proven effective with both small and large samples [71]. The choice to employ this method in the study was motivated by its orientation towards exploratory research and its adaptability to accommodate diverse sample sizes. The PLS path modeling approach was used to evaluate the measurement model and the structural model results, following a two-stage procedure [101]. The PLS path modeling rules encompass outer and inner calculation phases. The researchers emphasized the importance of satisfying certain non-parametric assessment criteria, such as construct reliability, outer loadings, indicator reliability, and average variance extracted [102]. In the realm of measurement assessment, construct validity is defined as "the extent to which an operationalization measures the concept it is supposed to measure" [103]. The structural model involved an evaluation of the $R^2$ measures and the magnitude and statistical significance of the path coefficients. This was accomplished by employing a bootstrapping procedure consisting of 5000 resamples [104]. The measurement and structural models were estimated using the PLS algorithm, along with the bootstrapping and blindfolding procedures, through SmartPLS software version 4.

### 4. Results

#### 4.1. Outer Model Estimation

To assess the study measurement (outer) models, several systems of measurement were inspected, following the suggestions of Hair et al. [104]. As illustrated in Table 2, the outer path loadings estimate for the reflective factors significantly surpassed the minimum threshold of 0.70. The calculated "Composite Reliability" (CR) values indicated high levels of internal reliability for all reflective factors. Moreover, the "Average Variance Extracted" (AVE) values, signaling convergent validity, exceeded the recommended minimum of 0.50, affirming convergent validity across all factors. Internal reliability, as measured by "Cronbach's Alpha" ($\alpha$), consistently exceeded the minimum threshold of 0.70, ensuring robustness and consistency in Table 2: the outer t-value was utilized to gauge the significance level of individual items. In the evaluation of discriminant validity, both the Fornell and Larcker criterion [105] and the cross-loading criterion were taken into consideration.

**Table 2.** Psychometric properties of the study measurement.

| Conructs | | α | CR | AVE | Loadings | T | VIF |
|---|---|---|---|---|---|---|---|
| Mental health disorders (Lovibond and Lovibond, 1995) [82] | | | | | | | |
| Depression | | 0.965 | 0.968 | 0.828 | | | |
| X1 * | "I was unable to feel any happiness or positive emotions". | | | | 0.915 | 75.121 | 4.569 |
| X2 | "I struggled to start tasks and take initiative". | | | | 0.907 | 57.382 | 4.151 |
| X3 | "I saw no reason to be optimistic or have hope". | | | | 0.915 | 57.713 | 4.762 |
| X4 | "I felt sad and depressed". | | | | 0.920 | 59.265 | 4.265 |
| X5 | "I didn't feel good about myself or my worth". | | | | 0.908 | 50.273 | 4.455 |
| X6 | "I was unable to get excited or passionate about anything". | | | | 0.904 | 49.704 | 4.182 |
| X7 | "I felt that life had no purpose or significance". | | | | 0.900 | 55.352 | 3.886 |
| Anxiety | | 0.956 | 0.963 | 0.792 | | | |
| X8 | "I noticed my mouth was dry". | | | | 0.868 | 61.218 | 3.961 |
| X9 | "I had trouble breathing (such as rapid or shortness of breath without exertion)". | | | | 0.901 | 93.160 | 4.552 |
| X10 | "I felt shaking in my hands". | | | | 0.870 | 66.811 | 3.764 |
| X11 | "I was concerned about potentially panicking and embarrassing myself". | | | | 0.877 | 73.648 | 4.569 |
| X12 | "I felt close to a panic attack". | | | | 0.908 | 94.754 | 4.265 |
| X13 | "I felt scared without a clear reason". | | | | 0.901 | 75.765 | 4.003 |
| X14 | "I became aware of my heartbeat without any physical activity (such as feeling my heart rate increase or skipping a beat)". | | | | 0.903 | 96.912 | 4.268 |
| Stress | | 0.948 | 0.960 | 0.761 | | | |
| X15 | "I had trouble calming down and relaxing". | | | | 0.821 | 35.456 | 2.459 |
| X16 | "I tended to have extreme reactions to events". | | | | 0.892 | 62.585 | 4.627 |
| X17 | "I felt I was employing a lot of nervous energy". | | | | 0.867 | 55.020 | 4.293 |
| X18 | "I became easily irritated". | | | | 0.890 | 65.880 | 3.983 |
| X19 | "I had difficulty finding relaxation". | | | | 0.885 | 55.035 | 3.682 |
| X20 | "I became easily frustrated by anything that disrupted my plans". | | | | 0.891 | 76.941 | 4.121 |
| X21 | "I felt I was easily upset or quick to anger". | | | | 0.859 | 48.103 | 3.573 |
| QOL (Diener et al., 1985) [83] | | 0.938 | 0.938 | 0.802 | | | |
| Y1 | "In most ways my life is ideal". | | | | 0.914 | 83.180 | 4.276 |
| Y2 | "I am satisfied with my life". | | | | 0.899 | 57.046 | 3.673 |
| Y3 | "The conditions of my life are excellent". | | | | 0.908 | 76.308 | 3.929 |
| Y4 | "So far, I have gotten the important things I want in life". | | | | 0.910 | 74.371 | 3.752 |
| Y5 | "If I could live my life over, I would change almost nothing". | | | | 0.844 | 41.624 | 2.307 |

**Table 2.** *Cont.*

| Conructs | | α | CR | AVE | Loadings | T | VIF |
|---|---|---|---|---|---|---|---|
| Religiosity (Lombardi et al., 2011) [84] | | 0.975 | 0.982 | 0.877 | | | |
| M1 | "I experience God's presence in my life". | | | | 0.890 | 67.377 | 3.492 |
| M2 | "My faith permeates my entire existence". | | | | 0.981 | 221.909 | 4.203 |
| M3 | "I am a religious one who allows religious considerations to impact my daily life | | | | 0.982 | 234.735 | 4.644 |
| M4 | "My religious beliefs are the foundation of my entire life philosophy". | | | | 0.981 | 232.826 | 2.112 |
| M5 | "Nothing is more important to me than serving God to the best of my ability". | | | | 0.733 | 4.365 | 1.812 |
| M6 | "When making important decisions, one should seek God's guidance". | | | | 0.984 | 260.789 | 3.162 |
| M7 | "I strive to incorporate my religion into every aspect of my life". | | | | 0.975 | 172.363 | 2.908 |
| Social connectedness Alsubaie et al. [31] | | 0.987 | 0.990 | 0.940 | | | |
| M8 | "I feel more comfortable when someone from my family is constantly with me". | | | | 0.982 | 195.092 | 2.279 |
| M9 | "My life is incomplete without a buddy beside me". | | | | 0.989 | 356.517 | 3.112 |
| M10 | "It's hard for me to use my skills and talents without someone beside me". | | | | 0.988 | 367.445 | 3.026 |
| M11 | "I stick to my friends and family like glue". | | | | 0.984 | 219.822 | 2.303 |
| M12 | "I join groups more for friendship than the activity itself". | | | | 0.987 | 335.613 | 3.886 |
| M13 | "I wish to find someone who can be with me all the time" | | | | 0.884 | 66.817 | 3.531 |

\* X1–X21: variables that measure mental health disorders and Y1–Y5: items that measure QOL. M1–M7: items that measure religiosity, M8–M13: items that measure social connectedness, CR: "composite reliability", AVE: "average variance extracted", and T: "T-value".

The off-diagonal values in Table 3: "Fornell and Larcker criterion" metrics for discriminant validity illustrate the correlation matrix among the study's latent factors. The shared scores between the constructs essentially denote squared correlations. A thorough examination of the loadings across the columns, as outlined in Table 4: "Loadings & Cross loadings scores", reveals a consistent pattern where the loadings of an indicator on its designated construct consistently exceed all the loadings on other constructs. This unequivocally establishes the presence of discriminant validity among all the constructs. Furthermore, as depicted in Table 4, the cross-loading values offer additional confirmation of sufficient discriminant validity. Each item shows a high correlation with its respective dimension compared to any other dimensions in the study.

**Table 3.** "Fornell and Larcker criterion" metrics for discriminant validity.

| | I | II | III | IV | V | VI |
|---|---|---|---|---|---|---|
| Anxiety | 0.890 | | | | | |
| Depression | 0.653 | 0.910 | | | | |
| Quality of Life | −0.460 | −0.406 | 0.895 | | | |
| Religiosity | 0.309 | 0.477 | 0.257 | 0.936 | | |
| Social Connectedness | 0.325 | 0.513 | 0.191 | 0.852 | 0.970 | |
| Stress | 0.603 | 0.339 | −0.359 | 0.103 | 0.158 | 0.873 |

**Table 4.** "Loadings & Cross loadings scores".

|  | Anxiety | Depression | Quality of Life | Religiosity | Social Connectedness | Stress |  |
|---|---|---|---|---|---|---|---|
| X1 * | 0.607 | 0.915 | −0.370 | 0.448 | 0.458 | 0.341 | 0.607 |
| X2 | 0.569 | 0.907 | −0.398 | 0.443 | 0.461 | 0.290 | 0.569 |
| X3 | 0.621 | 0.915 | −0.367 | 0.419 | 0.444 | 0.326 | 0.621 |
| X4 | 0.585 | 0.920 | −0.323 | 0.466 | 0.495 | 0.300 | 0.585 |
| X5 | 0.567 | 0.908 | −0.335 | 0.447 | 0.478 | 0.263 | 0.567 |
| X6 | 0.575 | 0.904 | −0.366 | 0.393 | 0.439 | 0.292 | 0.575 |
| X7 | 0.625 | 0.900 | −0.411 | 0.426 | 0.492 | 0.337 | 0.625 |
| X8 | 0.868 | 0.559 | −0.382 | 0.232 | 0.249 | 0.456 | 0.868 |
| X9 | 0.901 | 0.595 | −0.500 | 0.203 | 0.221 | 0.525 | 0.901 |
| X10 | 0.870 | 0.571 | −0.373 | 0.360 | 0.369 | 0.528 | 0.870 |
| X11 | 0.877 | 0.571 | −0.348 | 0.325 | 0.341 | 0.555 | 0.877 |
| X12 | 0.908 | 0.620 | −0.381 | 0.359 | 0.352 | 0.572 | 0.908 |
| X13 | 0.901 | 0.565 | −0.426 | 0.239 | 0.263 | 0.528 | 0.901 |
| X14 | 0.903 | 0.587 | −0.419 | 0.246 | 0.263 | 0.596 | 0.903 |
| Y1 | −0.370 | −0.351 | 0.914 | 0.238 | 0.182 | −0.282 | −0.370 |
| Y2 | −0.405 | −0.374 | 0.899 | 0.194 | 0.169 | −0.319 | −0.405 |
| Y3 | −0.404 | −0.351 | 0.908 | 0.228 | 0.174 | −0.294 | −0.404 |
| Y4 | −0.403 | −0.384 | 0.910 | 0.220 | 0.151 | −0.295 | −0.403 |
| Y5 | −0.465 | −0.356 | 0.844 | 0.267 | 0.179 | −0.408 | −0.465 |
| M1 | 0.261 | 0.454 | 0.280 | 0.890 | 0.838 | 0.075 | 0.261 |
| M2 | 0.307 | 0.461 | 0.242 | 0.981 | 0.823 | 0.104 | 0.307 |
| M3 | 0.307 | 0.472 | 0.250 | 0.982 | 0.823 | 0.091 | 0.307 |
| M4 | 0.307 | 0.460 | 0.247 | 0.981 | 0.826 | 0.105 | 0.307 |
| M5 | 0.250 | 0.325 | 0.188 | 0.733 | 0.582 | 0.099 | 0.250 |
| M6 | 0.300 | 0.459 | 0.241 | 0.984 | 0.826 | 0.106 | 0.300 |
| M7 | 0.292 | 0.472 | 0.220 | 0.975 | 0.821 | 0.099 | 0.292 |
| M8 | 0.301 | 0.500 | 0.164 | 0.829 | 0.982 | 0.148 | 0.301 |
| M9 | 0.305 | 0.488 | 0.193 | 0.831 | 0.989 | 0.151 | 0.305 |
| M10 | 0.322 | 0.490 | 0.198 | 0.826 | 0.988 | 0.164 | 0.322 |
| M11 | 0.323 | 0.500 | 0.189 | 0.832 | 0.984 | 0.142 | 0.323 |
| M12 | 0.326 | 0.486 | 0.192 | 0.824 | 0.987 | 0.164 | 0.326 |
| M13 | 0.311 | 0.524 | 0.173 | 0.820 | 0.884 | 0.149 | 0.311 |
| X15 | 0.636 | 0.422 | −0.335 | 0.160 | 0.214 | 0.821 | 0.636 |
| X16 | 0.490 | 0.251 | −0.282 | 0.089 | 0.109 | 0.892 | 0.490 |
| X17 | 0.395 | 0.193 | −0.224 | 0.042 | 0.086 | 0.867 | 0.395 |
| X18 | 0.552 | 0.335 | −0.294 | 0.155 | 0.200 | 0.890 | 0.552 |
| X19 | 0.515 | 0.287 | −0.277 | 0.119 | 0.175 | 0.885 | 0.515 |
| X20 | 0.578 | 0.316 | −0.403 | 0.038 | 0.095 | 0.891 | 0.578 |
| X21 | 0.454 | 0.218 | −0.318 | 0.029 | 0.083 | 0.859 | 0.454 |

* X1–X21: variables that measure mental health disorders and Y1–Y5: items that measure QOL. M1–M7: items that measure religiosity and M8–M13: items that measure social connectedness.

### 4.2. Structural (Inner) Model and Hypotheses Testing

After evaluating the model's predictive capacity and analyzing the strength and significance of paths across dimensions in accordance with the guidelines by Hair et al. [106], it is crucial to address the potential collinearity within the structural model before delving into the study hypotheses. In the realm of PLS path modeling, acknowledging multicollinearity is imperative for ensuring the results' validity [107].

To gauge collinearity, several factors were scrutinized. As depicted in Table 2, tolerance levels were below the VIF (Variance Inflation Factor) threshold of 5.00 for predictor dimensions, indicating the absence of significant collinearity. Subsequently, the significance of the estimates, the magnitude of the $R^2$ scores, the $f^2$ effect size, and predictive relevance metrics encompassing the $Q^2$ and $q^2$ effect sizes were meticulously examined.

We employed the PLS-SEM algorithm to calculate the significance of the paths in the structural model, estimating the path coefficients within the structural model to illustrate the envisioned connections between reflective factors. The bootstrapping option with 5000 resamples was applied to determine the significance of these path coefficients, and the detailed outcomes of hypothesis testing are shown Table 5: Hypotheses analysis. Furthermore, the PLS algorithm was utilized to compute the $R^2$ values for the endogenous latent unobserved factors. For the endogenous latent construct, QOL ($R^2$ = 0.592, $Q^2$ = 0.533), indicates a relatively high level of predictivity [108].

**Table 5.** Hypotheses analysis.

| Hypotheses | β | STDEV | T-Value | *p* | Conclusion |
|---|---|---|---|---|---|
| Depression → QOL | −0.29 | 0.06 | 4.40 | 0.000 | Supporting H1 |
| Anxiety → QOL | −0.23 | 0.076 | 3.101 | 0.002 | Supporting H2 |
| Stress → QOL | −0.12 | 0.063 | 1.982 | 0.048 | Supporting H3 |
| Moderating paths | | | | | |
| Social Connectedness x Depression → QOL | 0.030 | 0.155 | 0.197 | 0.844 | Rejecting H4 |
| Social Connectedness x Anxiety → QOL | −0.12 | 0.186 | 0.646 | 0.518 | Rejecting H5 |
| Social Connectedness x Stress → QOL | 0.263 | 0.120 | 2.198 | 0.028 | Supporting H6 |
| Religiosity x Depression → QOL | 0.099 | 0.158 | 0.628 | 0.530 | Rejecting H7 |
| Religiosity x Anxiety → QOL | 0.293 | 0.196 | 3.496 | 0.035 | Supporting H8 |
| Religiosity x Stress → QOL | −0.12 | 0.122 | 1.051 | 0.293 | Rejecting H9 |

The study assesses nine hypotheses, three direct and six moderations as illustrated in Figure 1. The evaluation process involved the use of bootstrapping with 5000 repetitions to calculate the critical parameters, including regression weights (β), t-values, and significance levels (*p*).

As depicted in Figure 1: PLS-SEM model output and outlined in detail in Table 5: Hypotheses analysis, the empirical findings extracted from the PLS-SEM output corroborate the validity of Hypothesis 1 (H1). This hypothesis suggested a negative correlation between depression, a facet of mental health disorders, and Quality of Life (QOL). The path analysis uncovered a significant negative effect, with a path coefficient of −0.29, a t-statistic of 4.40, and a *p* value below 0.001. Similarly, Hypothesis 2 (H2), proposing a detrimental impact of anxiety, another dimension of mental health disorders, on QOL, received support (β = −0.23, t = 3.101, *p* < 0.01). Additionally, Hypothesis 3 (H3) was substantiated, indicating an adverse and significant influence of stress, the third dimension of mental health disorders, on the QOL of individuals with disabilities. The path analysis yielded a path coefficient of −0.12, a t-statistic of 1.982, and a *p* value below 0.05.

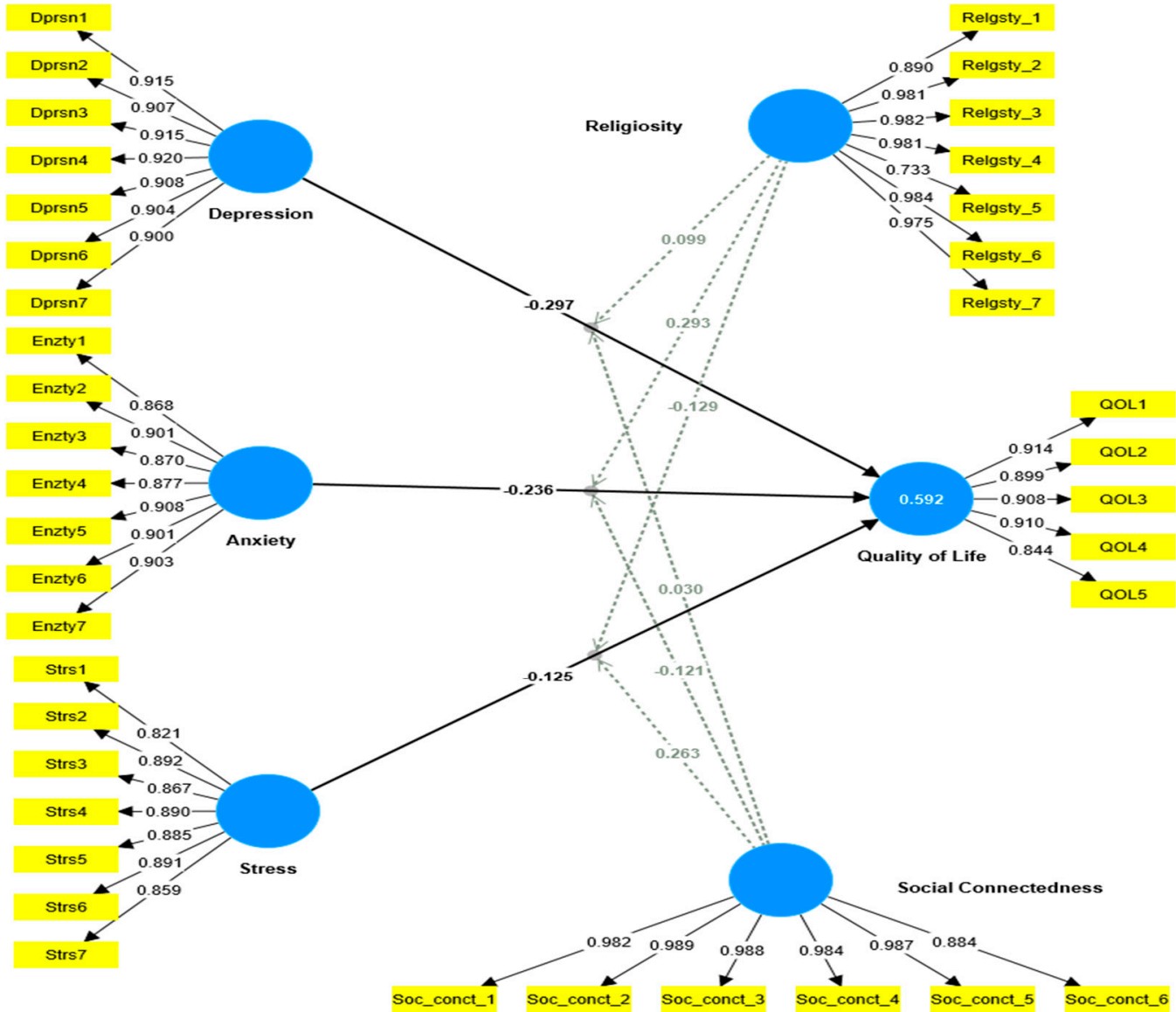

**Figure 1.** PLS-SEM model output.

In terms of the moderating analysis, the results obtained from the PLS-SEM output reveal that two out of the six moderating hypotheses were supported (as depicted in Figure 2). The negative impact of stress on QOL was dampened with the moderation effect of social connectedness (β = −0.263, t = 2.198, *p* < 0.05), supporting H6. Similarly, the negative impact of anxiety on QOL was dampened with the moderation effect of religiosity (β = 0.293, t = 3.496, *p* < 0.05), supporting H8. However, the findings, as depicted in Table 5: Hypotheses analysis, indicated that social connectedness failed to dampen the negative effect of depression (β = 0.030, t = 0.197, *p* = 0.844) and anxiety (β = −0.12, t = 0.646, *p* = 0.518) on QOL, which rejects H4 and H5. Similarly, the findings, as depicted in Table 5: Hypotheses analysis, revealed that religiosity failed to dampen the negative effect of depression (β = 0.099, t = 0.628, *p* = 0.530) and stress (β = −0.12, t = 1.105, *p* = 0.293) on QOL, which rejects H7 and H9. These outcomes are visually represented in Figure 2.

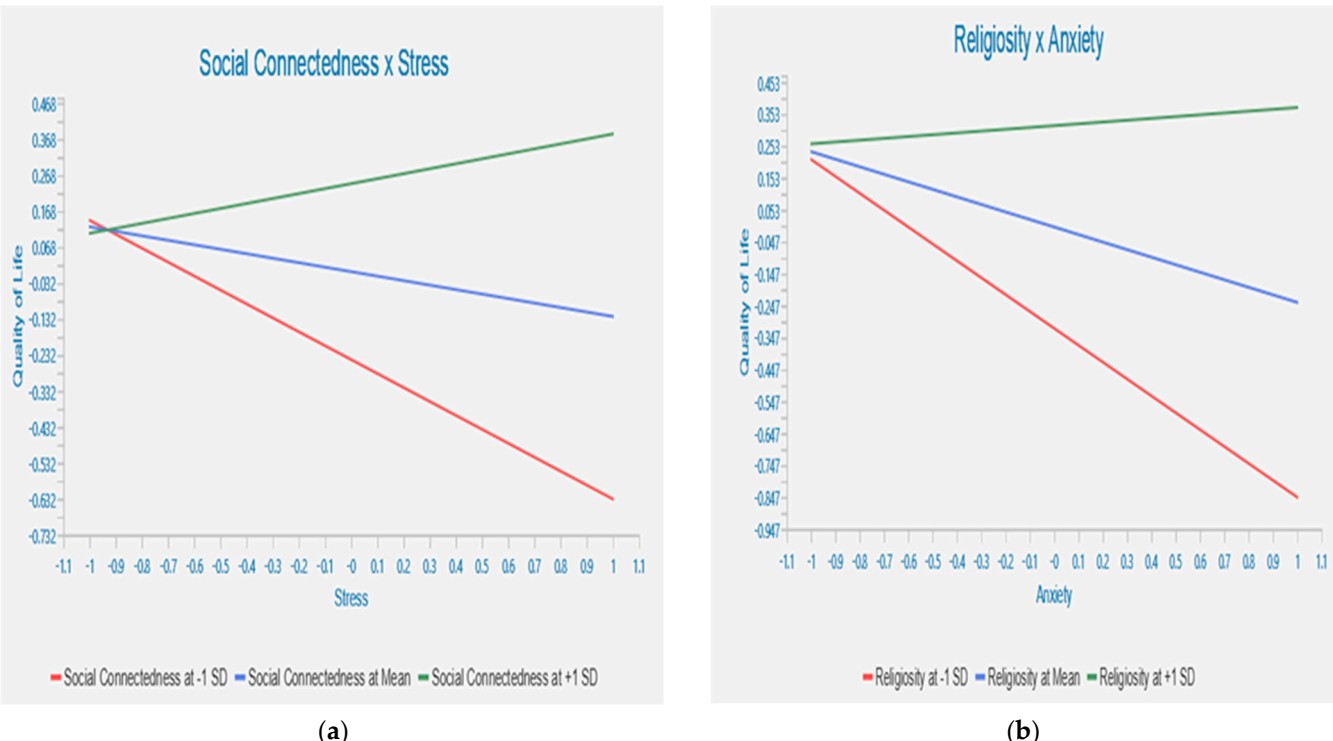

**Figure 2.** Slop analysis for the moderating effects. (**a**) Social connectedness x stress. (**b**) Religiosity x Anxiety.

## 5. Discussion and Conclusions

This study aimed to find out how sustainable MH relates to QOL. This research aimed to determine the important factors that influence the academic life quality of university students with impairments. The study examined the major opportunities and challenges that individuals with disabilities face while attending college.

This study focused on six aspects of relationships: (1) the direct association between stress, anxiety, depression, and QOL; and (2) the moderating effect of social connectedness and religiosity in the expected relationship between stress, anxiety, and depression on QOL. The results of this study supported the conclusions of previous research showing a negative correlation between depression, academic progress, and QOL [33,35]. University students with disabilities are more prone to experience depression than their peers without disabilities. Students with disabilities already face various obstacles in their educational journey. Due to their condition, students may experience additional difficulties in the classroom and in social situations. These difficulties may intensify the detrimental effects of depression, which could have a compounding effect on QOL and academic achievement.

Depression can further exacerbate these challenges, making it even more difficult to navigate the demands of schoolwork, social interactions, and extracurricular activities. In addition, because of their condition, students with disabilities could experience diminished self-esteem and self-efficacy. Furthermore, the stigma surrounding mental health can make it difficult for students with disabilities to seek help for depression. They may fear judgment, discrimination, or a lack of understanding from their peers, teachers, or support staff.

The findings of this study indicated that anxiety had a direct impact on QOL. These findings are consistent with other studies that suggest that anxiety has a negative impact on students' quality of life (QOL). According to a recent study, anxiety has a direct effect on QOL, which means that higher levels of anxiety are associated with a poorer QOL [37,109]. Anxiety can cause academic problems since it makes it difficult to focus and learn [49]. It can also lead to social separation, tiredness, and a general lack of vitality. Furthermore, the findings of the study revealed that anxiety has a considerable impact on QOL. Earlier

research findings reinforce this conclusion [110]. Furthermore, anxiety can cause a number of physical health problems, such as muscle strain.

Due to their disability, these students already encounter a variety of academic problems. These difficulties can be made worse by anxiety, which increases tension and makes it harder to focus, study, and finish tasks. Their QOL may be further impacted by receiving worse marks, missing deadlines, and academic dissatisfaction. Due to a lack of resources or stigma, students with disabilities may already encounter obstacles while trying to receive mental health help. Their dread, self-doubt, or negative thoughts can further impede their capacity to seek help as a result of anxiety. Their QOL may suffer, and their condition may become worse as a result of this lack of assistance.

According to our research findings, stress has a detrimental direct effect on QOL. This outcome was consistent with the research of other writers who found a connection between stress and the academic performance of college students with disabilities [62]. Stress can impair motivation, interfere with recall, and make focusing harder. It may also result in physical problems like headaches, stomachaches, and weariness, making it difficult to attend classes and participate in academic events.

This research finds that social connectedness moderates the relationship between stress and QOL. This result aligns with previous research findings [72]. This indicates the important role played by university disability service units and family connectedness, which are responsible for creating and sustaining a socialized learning environment for dampening the negative impact of stress on QOL among disabled people (specially students with disabilities). In addition, our findings show that religiosity moderates the relationship between anxiety and QOL. This result contributes to the research's findings in the context of disablists. Thus, the result also indicates the important role played by university disability service units, which are supposed to provide students with disabilities with non-class activities that are designed to enhance their religiosity level. Additionally, our findings show no moderating role played by social connectedness in improving the relationship between depression and QOL.

We can justify this result with a number of reasons. Individuals experience depression in many ways. Some persons may retreat from social engagements as a result of symptoms such as depression, anhedonia (lack of enjoyment), or exhaustion. Isolation can exacerbate their sadness and QOL, resulting in a negative feedback loop. Furthermore, those suffering from depression may be afraid of being stigmatized or judged when seeking social support [111]. This can make it difficult for individuals to reach out and obtain the benefits of social connection. Also, our findings show no moderating effect played by social connectedness in improving the relationship between anxiety, depression, and QOL.

The significant moderation effect of social connection on the relationship between stress and quality of life among students with disabilities can be justified by considering the role of social support as a coping mechanism. Stress, particularly in the context of disability, can significantly impact an individual's overall well-being [112]. Social connections act as a buffer against the negative effects of stress, providing emotional, instrumental, and informational support. Students with disabilities who experience higher levels of stress may find solace and assistance through their social networks, positively influencing their quality of life.

However, the absence of a significant moderation effect between anxiety or depression and quality of life by social connection may be explained by the complex and multifaceted nature of these mental health conditions. Anxiety and depression may involve internal factors, cognitive processes, and individual perceptions that are not as readily influenced by external social support. Individuals experiencing anxiety or depression may have diverse coping mechanisms, and the impact of social connection may vary among students. Moreover, these mental health conditions may require specialized interventions, such as counseling or therapeutic approaches, which social connections alone might not fully address.

Moreover, the study findings show no moderating effect played by religiosity on improving the relationship between depression and QOL, as well as the relationship between stress and QOL. The significant moderation effect of religiosity on the relationship between anxiety and quality of life among students with disabilities, as opposed to its lack of moderation for stress or depression, can be explained by considering the unique role of religiosity in addressing anxiety-related challenges. Religiosity often provides emotional support and coping mechanisms that are particularly effective in dealing with anxiety. Spiritual practices, beliefs, and community engagement may offer a sense of comfort, hope, and reassurance, positively impacting the emotional aspects of anxiety [113]. Anxiety is characterized by uncertainty and fear about the future. Religiosity, with its focus on faith and trust in a higher power, can help individuals to manage and reduce the uncertainty associated with anxiety.

In summary, this study aimed to investigate the primary function of mental health components in students with disabilities who aspire to become productive members of their community. Our findings confirm our main hypotheses (H1, H2, and H3) by demonstrating that all dimensions of mental health disorders—stress, depression, and anxiety—have a detrimental and significant impact on the quality of life of individuals with disabilities. Furthermore, the moderating effect of social connectedness mitigated the detrimental effects of stress on QOL (H4, H5, and H6). Furthermore, our research validated H6 by showing that social connectivity has a moderating influence on the association between stress and QOL. However, the association between anxiety and depression does not seem to be improved by social contact. We therefore dismiss H4 and H5. Likewise, the moderating effect of religiosity mitigated the detrimental effect of anxiety on QOL (H7, H8, and H9). Thus, our research validated H8 by demonstrating that religiosity has a moderating influence on the connection between anxiety and QOL. However, it does not seem like religiosity improves the connection between depression and stress. We therefore dismiss H7 and H9.

This study is also expected to have educational, social, health, and environmental implications. Few studies have been undertaken to determine the role of the main components of MH in students with disabilities. It intends to make significant contributions to the community and to support SDG3. Thus, the study's findings met its purpose by emphasizing the moderating impacts of religion and social connection on the relationship between mental health and the overall QOL experienced by students with disabilities.

## 6. Limitations and Further Study Opportunities

Similar to various studies in the social sciences, our research has certain limitations. The study's findings may be limited in terms of generalization, as it focuses on university students with disabilities. The specific nature and characteristics of the disability may influence the applicability of the results to a broader population. Future research could employ longitudinal designs to track changes in mental health and quality of life over time, providing a more dynamic understanding of these constructs among university students with disabilities. Another limitation is that it depends on participants' subjective assessments of their mental health and quality of life, which may introduce response bias. In this context, individuals might offer responses that align with societal expectations or demonstrate social desirability bias, potentially affecting the precision of the collected data. Additionally, limitations are present in the scales utilized to measure religiosity and social connectedness. The self-report nature of these scales relies on participants' perceptions of their religiosity and social connections, which may not fully encompass the richness and complexity of these constructs. Future research could incorporate more objective measures or combine self-reports with external assessments to enhance the robustness of the findings. Furthermore, the study primarily examines the moderating role of religiosity and social connectedness, leaving room for unexplored variables that could contribute to the mental health and quality of life outcomes in university students with disabilities. Investigating additional factors such as academic accommodations, socio-economic status, or the nature of the disability could provide a more comprehensive understanding of the

intricate interplay of variables affecting this population. Addressing these limitations in future research endeavors will contribute to a more nuanced understanding of the complexities associated with the well-being of this particular demographic.

## 7. Recommendations

In order to give students with disabilities a secure and accepting environment in which to interact, exchange stories, and speak up for their needs, universities must promote the establishment and upkeep of disability student unions. Disability student unions can provide peer mentorship programs, workshops, and social gatherings to help people feel like they belong. Universities should also make campus-wide events accessible to students with disabilities in order to promote involvement in groups, sports teams, and volunteer organizations. To guarantee equitable participation, the required accommodations must be made available, such as accessible locations and alternative material formats.

Universities should train instructors, staff, and student support providers to better understand disability challenges and foster inclusive interactions. A campus culture should be created that values diversity and prioritizes the needs of students with disabilities. Peer assistance programs that connect disabled pupils with non-disabled peers should be implemented. This can aid in the removal of obstacles, the promotion of mutual understanding, and the formation of genuine friendships. Peer mentors should be encouraged to offer emotional support, academic help, and advice on how to navigate school life.

This study has several positive implications, including social and academic implications. Decision-makers like the Deanship of Students Affairs, the Disability Services Unit, and the health authorities should improve the QOL for students with disabilities in different ways. Socially, the role expected to be played by disabled students can be developed when they have been given a chance to interact with their peers, friends, and local communities. By engaging students with disabilities in social events, society is advocating for inclusivity and treating all individuals with equal regard.

The process of socializing offers students with disabilities the chance to cultivate and exercise their social capabilities. Interacting with peers and partaking in collective endeavors assists them in establishing connections, refining their communication proficiencies, and augmenting their overall social welfare. Social endeavors play a vital role in the holistic welfare and caliber of existence for students with disabilities. Partaking in significant and pleasurable social exchanges can ameliorate psychological well-being, mitigate feelings of seclusion, and heighten overall contentment.

It is of critical significance to underscore that, when incorporating students with disability into social undertakings, it is imperative to guarantee accessibility, adaptability, and assistance in order to accommodate their distinct requirements and capabilities. Enhancing the lives of students with disabilities can yield significant academic implications and advantages. Some of the notable academic implications encompass the promotion of inclusive education.

By granting students with disabilities equal access to education, inclusive classrooms are established, thereby fostering an environment of diversity, comprehension, tolerance, and respect among students. This inclusive approach enables students with disabilities to actively engage in academic activities and acquire knowledge alongside their peers. Moreover, the enhancement of students with disabilities' lives facilitates the provision of equitable learning opportunities. This entails ensuring that students with disabilities have access to educational resources, support services, and assistive technologies tailored to their specific requirements.

By doing so, the aim is to level the educational playing field and guarantee that students with disabilities receive identical educational opportunities to their non-disabled counterparts. As a result, students with disabilities can realize their full academic potential. Furthermore, prioritizing the improvement of students with disabilities' lives within schools and educational institutions has positive outcomes in the social, economic, and health dimensions.

Collaboration among university and community stakeholders should be encouraged to ensure a complete and coordinated strategy for serving impaired students. Including impaired students in the conception and execution of new initiatives should be considered. In addition, the effectiveness of implemented recommendations should be monitored continuously and changed as needed based on feedback and statistics. This will ensure that interventions and support systems genuinely impact impaired students' academic and social experiences.

**Author Contributions:** Methodology, E.A.A.-S. and M.A.M.; Software, M.E.; Validation, M.M.A.; Formal analysis, M.A.Z.; Investigation, E.A.A.-S.; Data curation, M.A.Z.; Writing—original draft, M.E. and M.A.M.; Writing—review & editing, M.M.A., M.E. and M.A.M.; Visualization, M.E.; Supervision, E.A.A.-S.; Project administration, E.A.A.-S.; Funding acquisition, M.A.Z. All authors have read and agreed to the published version of the manuscript.

**Funding:** The authors extend their appreciation to the King Salman Center for Disability Research for funding this work through Research Group no KSRG-2023-534.

**Institutional Review Board Statement:** The study was conducted according to the guidelines of the Declaration of Helsinki and approved by the Deanship of Scientific Research Ethical Committee, King Faisal University (project number: KFU-REC-2023-AUG-ETHICS1064, date of approval: 6 September 2023).

**Informed Consent Statement:** Informed consent was obtained from all subjects involved in the study.

**Data Availability Statement:** Data are available upon request. Kindly contact the first author privately through e-mail.

**Conflicts of Interest:** The authors declare no conflict of interest.

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
