# Peer review of "Mental Health and Quality of Life among University Students with Disabilities: The Moderating Role of Religiosity and Social Connectedness"

_sustainability, doi:10.3390/su16020644_

Round 1
Reviewer 1 Report
Comments and Suggestions for Authors
Title and Abstract Critique:
Title:
The title succinctly captures the main variables under investigation: "Mental Health and Quality of Life among Disabled Students: Religiosity as A Moderator and Social Connectedness as A Mediating." However, it could benefit from greater clarity and specificity regarding the scope of disabilities and the student population studied.
Abstract:
Strengths:
· Clear Objective: The abstract outlines the primary aim of the study, focusing on the relationships between mental health, quality of life, religiosity, and social connectedness among individuals with disabilities.
· Methodology Summary: It briefly explains the methodology employed (Partial Least Squares Structural Equation Modeling) to explore the relationships and effects within the proposed framework.
· Key Findings: It highlights the impact of mental health dimensions on the quality of life and the moderating effects of social connectedness and religiosity, offering specific insights into the moderation effects of these factors on stress and anxiety.
Introduction Critique:
Strengths:
· Scope Identification: The introduction effectively highlights the increasing number of students with disabilities worldwide and their challenges in educational settings, setting the stage for the study's relevance.
· Clear Objectives: It clearly outlines the aim of the research in exploring the interplay between mental health (MH), quality of life (QOL), religiosity, and social connectedness among disabled students, providing a clear context for the study.
· Reference Integration: It incorporates references to support various claims, providing a foundation for the study and demonstrating existing literature gaps.
Suggestions for Improvement:
· Structure and Flow: The introduction lacks a clear structure, leading to a somewhat disjointed flow of ideas. Consider restructuring paragraphs to create a smoother transition between points, making it easier for readers to follow the logical progression of the narrative.
· Clarification of Research Gap: While the introduction hints at the gap in the literature related to the moderating role of religiosity and social connectedness, it could explicitly articulate how this study contributes to filling that gap and why it's essential.
· Simplification and Conciseness: Some sentences are overly complex or repetitive, which may hinder readability. Simplify sentences where possible and avoid repeating ideas across paragraphs.
Overall, the introduction effectively introduces the research topic and its significance but could benefit from a clearer structure, more explicit identification of the research gap, and improved readability through simplification and conciseness.
Methods Critique:
Strengths:
· Clear Description of Methodology: The section provides a detailed overview of the study's quantitative approach, questionnaire design, and data collection procedures, enhancing transparency and reproducibility.
· Justification of Measurement Tools: The rationale behind the selection of measurement scales (e.g., DASS-21, SWLS, religiosity, social connectedness) is well-articulated, indicating their appropriateness for assessing the intended constructs.
· Rigorous Data Handling and Addressing Bias: The section shows a meticulous approach to handling missing data, addressing common method variance (CMV), and assessing non-response bias, thereby ensuring data reliability.
Suggestions for Improvement:
· Clarity and Simplification: Some sections could benefit from clearer explanations and simplification of technical terms or methodologies. This could enhance readability and understanding, especially for non-experts in the field.
· Structural Organization: Consider breaking down lengthy paragraphs into smaller, more digestible sections. Clear subheadings could help readers navigate through different aspects of the methodology more easily.
· Emphasis on Limitations: While the section outlines data collection and analysis methods comprehensively, including a brief discussion of potential limitations (e.g., limitations of the chosen scales, biases despite attempted controls) would add depth to the methodological rigor.
Overall, the methods section is thorough and well-structured, but minor improvements in clarity, organization, and an explicit mention of potential limitations could enhance the overall strength of the methodology.
The Results section of the scientific article appears well-structured and comprehensive in reporting the outcomes of the study. Here's a critical assessment with some suggestions for improvement:
Strengths:
· Rigorous Analysis: The section demonstrates a comprehensive analysis of the measurement (outer model) and the structural (inner model) aspects of the study, utilizing various statistical techniques, such as loadings, CR, AVE, T-values, VIF, and path coefficients.
· Validation of Constructs: The authors appropriately validate the constructs used in the study, ensuring reliability, convergent validity, and discriminant validity. The application of Fornell and Larcker criterion and cross-loadings helps confirm the distinctiveness of the constructs.
· Hypotheses Testing: The study rigorously tests multiple hypotheses, providing detailed statistical values, including beta coefficients, standard errors, t-values, and P-values, to support or reject each hypothesis. The interpretation of these results is clear and tied back to the research questions.
Areas for Improvement:
· Clarity in Presentation: While the analysis is thorough, the presentation could be more reader-friendly. Breaking down complex tables into smaller, digestible parts or providing clearer section headers within tables could aid in understanding for readers not deeply familiar with statistical analyses.
· Explanation of Findings: While the statistical results are well-reported, a brief interpretation or discussion alongside each significant finding could enhance the understanding of the practical implications of the results.
The "Discussion and Conclusions" section offers valuable insights into the impact of various factors on the quality of life (QOL) for university students with impairments. Here's a critical assessment and potential areas for improvement:
Strengths:
· Alignment with Previous Research: The discussion aligns the current findings with existing literature, demonstrating a thorough understanding of prior research. This helps in establishing continuity and validity in the research area.
· Practical Implications: The section effectively outlines recommendations based on the study's findings, suggesting actionable steps for universities and stakeholders to improve the academic and social experiences of disabled students.
Areas for Improvement:
· Discussion Structure: The section lacks a clear delineation of sub-sections, which could make it easier to navigate and comprehend. Breaking down the discussion into subheadings corresponding to each research aspect (e.g., depression, anxiety, stress, moderating effects) could enhance readability.
· Interpretation of Findings: While the section presents the study's results, it lacks in-depth interpretation and contextualization of the findings. Adding explanations that relate the results to the broader context of disability in academia or society could enhance the discussion's depth.
· Acknowledgment of Limitations: Including a brief section highlighting the limitations of the study, such as sample size constraints or methodological limitations, would strengthen the article's credibility by acknowledging potential weaknesses.
Overall, the section effectively presents the study's findings and suggests practical recommendations. Enhancements in structuring the discussion, deeper interpretation of results, and acknowledgment of limitations could further strengthen the article's impact and credibility.
Author Response
Dear Reviewer,
Thank you for allowing us to resubmit a revised draft of our manuscript titled “Mental Health and Quality of Life Among University Students with Disabilities: The Moderating Role of Religiosity and Social Connectedness.” to “Sustainability Journal”.
We appreciate the time and effort that you have dedicated to providing your valuable feedback on our manuscript. We are grateful for your insightful comments on our paper. We have been able to incorporate changes to reflect the suggestions provided in our revised manuscript. We have marked the changes in red color in the manuscript.
Attached is a point-by-point response to the reviewer's comments and concerns.

Reviewer 2 Report
Comments and Suggestions for Authors
The authors pointed out the important issue of supporting students with disability. In order to improve their functioning at universities, the conclusions indicate the need to train university employees on the specific needs of people with disability. This is undoubtedly consistent with research results confirming that knowledge about the specific needs increases openness to people with disability in public space.
I find a number of shortcomings in the article that must be removed before its publication. I grouped the issues requiring correction as follows:
1. Purpose of the article.
In my opinion, the article should have a clearly defined purpose. Meanwhile, in the introduction, the authors focused on a broad analysis of the topic without indicating the actual purpose of the publication. This goal is probably to verify numerous hypotheses, but this is not clearly indicated in the text. I also do not know how the hypotheses relate to the specificity of students with disability.
2. Weaknesses regarding the research sample.
I consider this aspect to be the weakest in the presented article. There is no information about the number of universities to which a request to disseminate information about the study was sent, nor is there any information about the specific nature of the universities or their locations. There is no information on how respondents were selected. There is no information about who could be a potential respondent and how they were defined (screening questions). No information about the obtained research sample - to whom the questionnaires were ultimately addressed. The authors definitely miscalculated the response rate by calculating the percentage of fully completed questionnaires in relation to all received ones. Meanwhile, the response rate measures the response rate of the study, i.e. the ratio of questionnaires received to all sent ones, which cannot be calculated in the article due to the lack of data.
3. Corrections regarding the so-called technical nature, but very important for the quality of the article
In this area, it should be noted that there is a need to correct the terminology used by the authors. Various terms for people with disability appear in different parts of the article. We can find e.g. disabled students, disability student, students with impairments, etc. Meanwhile, the WHO recommends using the term disability as appropriate and correct with the current understanding of the issue of disability. More:
https://www.who.int/news-room/questions-and-answers/item/people-with-disability-vs-persons-with-disabilities
The authors also use numerous abbreviations, such as MH, QOL, QO, which makes reading the text very difficult, especially since these abbreviations already appear in the abstract without prior explanation. I suggest giving up shortcuts.
4. Editorial errors.
Missing title for Figure 3 and incorrect place of reference. Incomprehensible titles of tables 2 and 3. Unnecessary dots after the titles.
Author Response
Please find the response in the attached file. Thank you!

Reviewer 3 Report
Comments and Suggestions for Authors
The manuscript describes a quantitative study investigating correlations of depression, anxiety, stress and QoL, as well as the moderating effects of social connectedness and religiosity. The authors have made some comprehensive details in the analysis of results. The current work comes as a timely contribution with emerging studies focusing on the mental health of disabled students. This is also an important topic to be considered in the attainment of sustainable development. The value of the current work is supported and I have some comments below for enhancement of the study.
1.
The literature review and background sections can be strengthened. Section 1 Introduction can be added with descriptions and explanations of the importance of relevant and current studies in advancing associated SDGs as mental health of students is an importance issue in sustainable development and provision of quality education. A strengthened linkage between the current study and attainment of relevant SDG would benefit the support of current study. In addition, at the later part of the introduction / literature review, the authors would consider providing more details and more explicitly explain why specifically religiosity and social connectedness, amid other important items, are focused. The research questions and significance of the current study should be more detailed.
2.
Regarding the methods section on page 5-7:
Ln 232-233, are all the internal consistency given and elaborated for the different scales used? Any rationale for the basis of the sample size? Are there any inclusion / exclusion criteria for the invitation of participating the study? Ln 240-255, suggested also presenting the demographics in table.
3.
Ln 389-390, “the purpose of this research is to determine the important factors that influence the academic life quality of university students with impairments.” Are the result analysis in the previous result section considering all the participants? According to the descriptions on page 6 in the methods section, there were participants who are considered younger students (below 20 year-old), and are they all university students? Any demographics on their academic discipline / major of study in university, study year, etc.?
4.
It is suggested to include in a separate section on any limitations and bias of the current work and future extended work based on the current context.
5.
Editing or thorough checking is suggested. There are grammatical or writing issues observed that would have obscured the delivery and readability. For example, ln 39, can impacts; ln 99, “famil”; ln 432, “our find shows no…”; abstract, people) moreover.
Thank you for considering the details.
Author Response

(The authors gave the same response as above.)

Reviewer 4 Report
Comments and Suggestions for Authors
The following proposals are made:
a) Modify the title. The analyzes carried out are moderation, no mediation analysis is carried out.
b) Show the questions used in the questionnaire and, if they come from an already validated scale or questionnaire, identify it.
c) In the presentation of results, indicate the B0 ( intercepts ).
d) In the discussion, try to justify that social connection significantly moderates the relationship between stress and quality of life, and does not moderate the relationship between anxiety and quality of life, and that religiosity significantly moderates the relationship between anxiety and quality of life, and does not moderate the relationship between stress and quality of life.

Author Response

(The authors gave the same response as above.)

Round 2
Reviewer 1 Report
Comments and Suggestions for Authors
The authors have made changes that improve the quality of the article, congratulations.
Author Response
Thank you!
Reviewer 2 Report
Comments and Suggestions for Authors
According to the WHO standard, it is recommended to use the singular in the case of disability. So students with disability and not students with disabilities. A disability is the same feature as eye color or hair color. Please adapt your terminology to this standard.
Author Response
Dear Reviewer
Thanks again for your efforts to improve the quality of our manuscript; we revised our manuscript according to your suggestion, and according to WHO standards, we have used the singular in the case of disability. So, it is students with disabilities and not students with disabilities.
Thanks again for all your constructive feedback